# Selective hydrodeoxygenation of α, β-unsaturated carbonyl compounds to alkenes

Tianjiao Wang[1,2], Yu Xin[1,2], Bingfeng Chen[1], Bin Zhang[1], Sen Luan[1,2], Minghua Dong[1,2], Yuxuan Wu[1,2], Xiaomeng Cheng[1,2], Ye Liu[1], Huizhen Liu [1,2] ✉ & Buxing Han [1,2]

Achieving selective hydrodeoxygenation of α, β-unsaturated carbonyl groups to alkenes poses a substantial challenge due to the presence of multiple functional groups. In this study, we develop a ZnNC-X catalyst (X represents the calcination temperature) that incorporates both Lewis acidic-basic sites and Zn-N$_x$ sites to address this challenge. Among the catalyst variants, ZnNC-900 catalyst exhibits impressive selectivity for alkenes in the hydrodeoxygenation of α, β-unsaturated carbonyl compounds, achieving up to 94.8% selectivity. Through comprehensive mechanism investigations and catalyst characterization, we identify the Lewis acidic-basic sites as responsible for the selective hydrogenation of C=O bonds, while the Zn-N$_x$ sites facilitate the subsequent selective hydrodeoxygenation step. Furthermore, ZnNC-900 catalyst displays broad applicability across a diverse range of unsaturated carbonyl compounds. These findings not only offer valuable insights into the design of effective catalysts for controlling alkene selectivity but also extend the scope of sustainable transformations in synthetic chemistry.

Olefins are versatile compounds used in coatings, plasticizers, organic synthesis, and polymerisation[1–5]. Although biomass is an important renewable carbon source for producing chemicals or fuels, reports on synthesising olefins via hydrodeoxygenation from biomass are limited[6–10]. α, β-unsaturated carbonyl compounds are important biomass derivatives. One potential method involves selectively breaking the C–O bond in α, β-unsaturated carbonyl compounds to preserve the C=C bond, but this is challenging due to the difficulty of selectively hydrodeoxygenation instead of reducing C=C double bonds.

Several literature sources have documented the synthesis of alkenes from α-β-unsaturated carbonyl compounds[11,12]. For instance, this has been achieved through reduction via hydrazone intermediates in the Wolff-Kishner-Huang reduction[13] or through xanthate intermediates in the Barton-McCombie reaction[14]. Cook et al. reported its accomplishment using a Ni(II) pre-catalyst and a silane-reducing agent[15], while Gómez et al. demonstrated the process in two reaction steps[16]. However, the reagents used in these methods are not considered safe or environmentally friendly. To facilitate one-pot cascade reactions, the development of bifunctional catalysts is essential. Various nanostructured materials such as multicompartmentalized mesoporous organosilicas[17], nanotubes[18], and hierarchical architectures[19] have been explored to fabricate efficient cascade catalysts. Bifunctional catalysts can be created by loading bimetallic components onto these nanostructured materials, thereby enabling effective catalysis through synergistic interactions between multiple sites.

Catalytic transfer hydrogenation (CTH) using low-cost and readily available alcohol compounds as an H-donor has gained attention due to its greener, more effective, and safer hydrogenation process with higher selectivity than H$_2$ as the reducing agent[20–24]. CTH finds

[1]Beijing National Laboratory for Molecular Sciences, CAS Key Laboratory of Colloid and Interface and Thermodynamics, CAS Research/Education Center for Excellence in Molecular Sciences, Center for Carbon Neutral Chemistry, Institute of Chemistry, Chinese Academy of Sciences, Beijing 100190, China. [2]School of Chemistry and Chemical Engineering, University of Chinese Academy of Sciences, Beijing 100049, China. ✉e-mail: liuhz@iccas.ac.cn

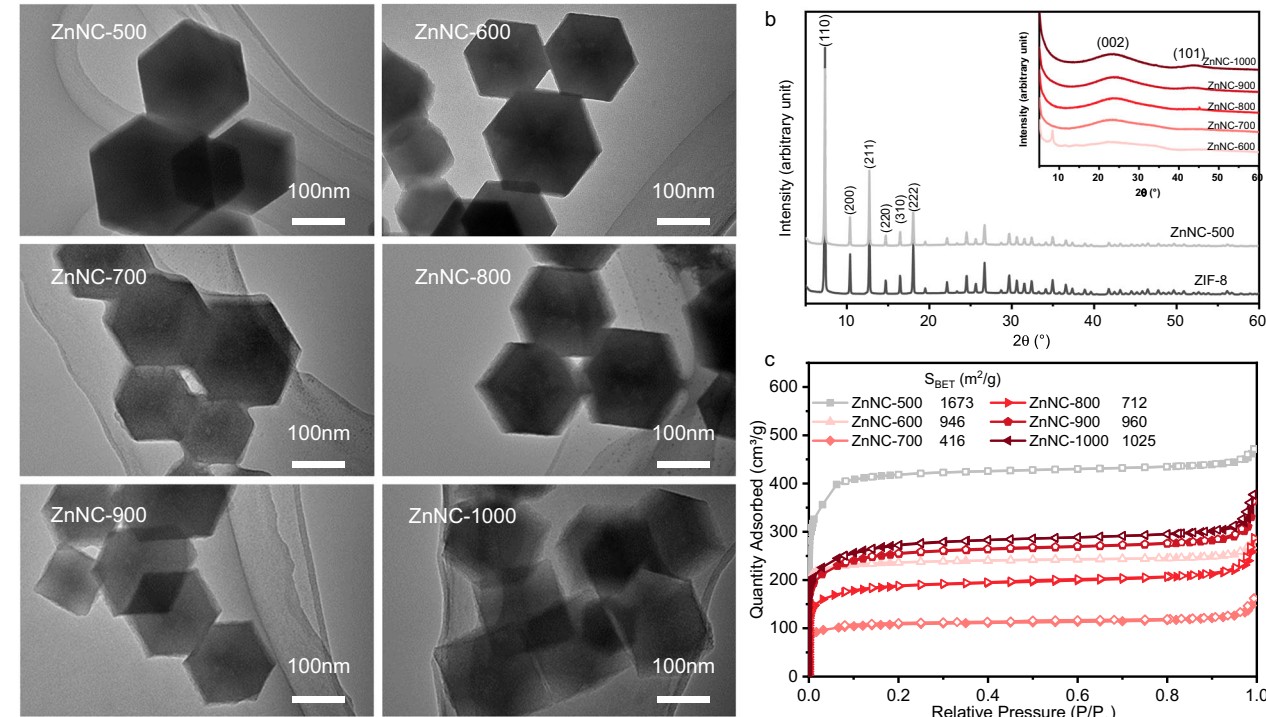

**Fig. 1 | The structural characterization of ZnNC-X catalysts. a** TEM images of ZnNC-X catalysts. **b** XRD patterns of ZnNC-X catalysts and ZIF-8. **c** N$_2$ adsorption-desorption isotherms.

widespread application in hydrodeoxygenation-based biomass upgrading[20–26]. Various Lewis acid-base catalysts have shown excellent CTH performance[23,26–28]. Metal-N-C materials, which exhibit Lewis acid behaviour at metal sites with positive valence and Lewis base characteristics at N sites, show promise as CTH catalysts[23,29,30]. Furthermore, these materials also use non-precious metal centres with low metal loadings, resulting in reduced costs and metal consumption, making them an ideal catalytic material.

Zinc, which is abundant, safe, and inexpensive, has been utilised in the development of Lewis acid catalysts for activating electron-rich groups such as alkyne and nitroarenes[31–33]. In addition, Zn-based catalysts with single Zn sites have been reported for reductive N-formylation of nitroarenes[34] and oxidative cleavage of C(CO)−C bonds[35], with single Zn acting as the active sites for these processes. However, their performance as CTH catalysts remains uninvestigated despite being used in different catalytic processes. In this work, we achieved the selective hydrogenolysis of α, β-unsaturated carbonyl compounds for the synthesis of olefinic compounds. This was accomplished by carefully controlling the acidity and basicity of the Zn-N-C catalyst surface. The reaction process involved two steps: first, hydrogenation of α, β-unsaturated carbonyl compounds to produce unsaturated alcohols, followed by hydrodeoxygenation of unsaturated alcohols to form olefins. Based on experimental data and catalyst characterization, it was found that the active sites for the first step reaction were the Lewis acid-base sites on the surface of ZnNC materials, while the active sites for the second step reaction were the Zn-N$_x$ species.

## Results

### Synthesis and characterization

ZnNC-X catalysts were synthesised via the pyrolysis of zeolitic imidazolate framework-8 (ZIF-8) under an Ar atmosphere (Supplementary Fig. 1), where X represents the calcination temperature. The morphology for catalysts was initially examined, and both scanning electron microscopy (SEM) and transmission electron microscopy (TEM)

images demonstrated that the catalysts maintained their original rhomboidal dodecahedron shape, even after being calcined at different temperatures (Fig. 1a; Supplementary Figs. 2–4). The catalyst's element composition was determined using inductively coupled plasma optical emission spectrometer (ICP-OES) and organic element analyser, as shown in Supplementary Table 1. The zinc content decreased with increasing calcination temperature. Powder X-ray diffraction (XRD) patterns were utilised to analyse the structure of catalysts (Fig. 1b). The diffraction peaks of the ZIF-8 parent material were consistent with prior research[36], indicating excellent crystallinity. The diffraction peaks of the ZnNC-500 catalyst were also comparable to those of ZIF-8, indicating that the crystal structure remained intact after calcination at 500 °C. The ZnNC-X catalysts (X = 600–1000) displayed two broad peaks at approximately 24° and 43°, corresponding to the (002) and (101) planes of graphitic carbon[35]. This suggests that the crystal structure of ZIF-8 was destroyed as the calcination temperature increased to above 600 °C. No characteristic peak of zinc metal or zinc oxide was detected in the XRD patterns (Supplementary Fig. 5). The N$_2$ adsorption-desorption isotherms indicated that the ZnNC-X catalysts had a higher BET surface area, and their curves were consistent with type I isotherm curves, indicating the presence of micropores (Fig. 1c).

### Catalytic performance

The catalytic performance of the catalysts was assessed for the hydrodeoxygenation of α, β-unsaturated carbonyl compounds to alkenes using cinnamaldehyde as a model compound (Fig. 2). The reaction pathway is illustrated in Figure 2B, where cinnamaldehyde (**1a**) is selectively converted to hydrocinnamaldehyde (**1b** HCAL) or cinnamyl alcohol (**1c** COL) via C=C or C=O addition. HCAL may be further hydrogenated to produce hydrocinnamyl alcohol (**1d** HCOL), while COL may also undergo hydrodeoxygenation to yield 1-propenylbenzene (**1e**) and allylbenzene (**1f**).

Table 1 shows that ZnNC-900 catalyst exhibited the highest catalytic activity for the hydrodeoxygenation of CAL to alkenes, with total

**Fig. 2 | Possible reaction pathways. A** Reaction pathway of α, β-unsaturated carbonyl compounds (R$_1$ = Ar, furyl; R$_2$ = Ar, H, CH$_3$). **B** Reaction pathway of cinnamaldehyde. **1a** cinnamaldehyde (CAL), **1b** hydrocinnamaldehyde (HCAL), **1c** cinnamyl alcohol (COL), **1d** hydrocinnamyl alcohol (HCOL), **1e** 1-propenylbenzene, **1f** alylbenzene, **1g** 1-propylbenzene.

yields of **1e** and **1f** reaching 61.1% (Table 1, Entry 5). Optimal reaction conditions were also determined, with no **1e** and **1f** detected at 90 °C and 120 °C (Supplementary Fig. 6). When the temperature was increased to 150 °C, **1e** and **1f** were detected, and the selectivity of alkenes increased significantly from 150 °C to 180 °C. These results suggest that below 150 °C, the primary reaction was hydrogenation of CAL to COL, while at 180 °C, the main reaction was the hydrodeoxygenation of COL to alkenes.

In order to obtain a better understanding of the hydrodeoxygenation of **1a** to alkenes, the two-step reaction was studied separately. The first step involves selective hydrogenation of **1a** to **1c**, while the second step is the hydrodeoxygenation of **1c** to **1e** and **1f**. Various catalysts were evaluated for the first step and, as shown in Table 2, ZIF-8 exhibited very low activity with only a 12.5% yield product (Table 2, Entry 2). For the catalysts prepared at different pyrolysis temperatures, the catalytic activity initially increases with increasing pyrolysis temperature, followed by a decrease, and then another increase. ZnNC-600 and ZnNC-900 catalysts demonstrated good catalytic performance for the selective hydrogenation of **1a** to **1c**, whereas different results were observed for the second step, and the activity of the catalysts increased with increasing pyrolysis temperature. Among the tested catalysts, ZnNC-900 catalyst showed the best catalytic performance (Table 2, Entry 7). Notably, Zn(NO$_3$)$_2$ and ZnO exhibited very low activity for both the first and second steps, indicating that Zn$^{2+}$ may not be the active site for the reaction (Table 2, Entry 9, 10). Furthermore, it was established that the presence of zinc is essential for the reaction, as evidenced by the use of the NC-900 catalyst without zinc, which resulted in no product formation (Table 2, Entry 11). Catalysts calcined at different temperatures had varying activity for the two steps, indicating different active sites for each reaction.

## Table 1 | Selective hydrodeoxygenation of cinnamaldehyde over various catalysts

| Entry | Catalysts | Conv. (%) | Y. (%) | | | |
|-------|-----------|-----------|--------|------|---------|------|
| | | | 1c | 1d | 1e + 1f | 1g |
| 1 | ZnNC-500 | 64.4 | 46.0 | – | – | 2.6 |
| 2 | ZnNC-600 | 99.9 | 80.4 | – | 2.2 | 6.4 |
| 3 | ZnNC-700 | 92.6 | 37.1 | – | 3.3 | 3.9 |
| 4 | ZnNC-800 | 99.9 | 4.5 | 9.2 | 48.0 | – |
| 5 | ZnNC-900 | 99.9 | 5.6 | 10.9 | 61.1 | 2.5 |
| 6 | ZnNC-1000 | 99.9 | 15.1 | 6.8 | 51.0 | – |

Reaction conditions: cinnamaldehyde (0.2 mmol), catalyst (20 mg), 2-propanol (4 mL), N$_2$ (1 MPa), 180 °C, 24 h. The conversion of substrate and the yield of products were determined by GC with dodecane as-internal standard.

## Reaction mechanism

CTH process of α, β-unsaturated aldehyde to α, β-unsaturated alcohol has been reported using alcohol as solvent and hydrogen donor and Lewis acid-base sites may be the active sites. Pyridine Fourier transform infrared spectroscopy (pyridine-FTIR) and CO$_2$ temperature programmed desorption (CO$_2$-TPD) were used to determine the acidity and basicity of the catalysts. Pyridine-FTIR showed that all ZnNC-X catalysts contained acid sites with a characteristic absorption peak of pyridine chemical adsorption by Lewis acidic site at 1446 cm$^{-1}$ (Supplementary Fig. 13)[37]. Although the peaks at 1540 cm$^{-1}$ assigned to the adsorption of pyridine to Brönsted acid site were also detected, its density was much lower than that of Lewis acidic site (Supplementary Table 3). The conversion of CAL selective hydrogenation to COL had the same trend as the Lewis acid density in the catalysts (Fig. 3), indicating that the activity of ZnNC-X catalysts correlated with the amounts of weak and medium-strong Lewis acids. Furthermore, low activity was observed in ZnNC-500 due to the absence of a basic site (Supplementary Fig. 12). These results suggest that the active site for the catalytic selective hydrogenation was a Lewis acid-base site.

Deuterium experiments were conducted to further determine the catalytic mechanism. A solvent mixture of 2-propanol-$d_8$ and $t$-butanol was used, with $t$-butanol excluded as a hydrogen source due to the absence of β-H .H/D exchange between the hydroxyl groups of $t$-butanol and 2-propanol-$d_8$ resulted in the conversion of most of the latter to 2-propanol-$d_7$ (Fig. 3c)[38]. COL produced from CAL hydrogenation was analysed by mass spectrometry and showed a 1 amu mass shift of the parent ion (from 134 to 135 amu, Fig. 3b), indicating the intermolecular hydride transfer mechanism. This means that β-D from 2-propanol-$d_7$ was directly transferred to CAL via a six-membered ring intermediate through Meerwein-Ponndorf-Verley (MPV) reduction[38,39] (Fig. 3c). Transfer of β-D from 2-propanol-$d_7$ to α-C of CAL was further confirmed by $^1$H NMR (Supplementary Fig. 14).

Infrared spectroscopy is an important technique for studying reaction pathways and surface adsorption of substrates and intermediates. In situ Fourier transform infrared spectroscopy (FTIR) experiments were used to investigate the adsorption of CAL and 2-propanol (IPA) on ZnNC-600 catalyst surface (Fig. 4). Pure CAL was introduced into the reaction cell at 40 °C for 20 min, followed by vacuuming to remove gaseous CAL and physically adsorbed species on the catalyst. The peaks observed at 1714 cm$^{-1}$ and 1205 cm$^{-1}$ correspond to the stretching vibration of the C = O bond and the rocking vibration of the aldehyde group's C−H bond, respectively (Fig. 4a)[40]. Over time, vacuuming caused a redshift of $v$(C = O) from 1714 cm$^{-1}$ to 1710 cm$^{-1}$, indicating the chemisorption of C = O in CAL on the catalyst. Subsequently, introducing IPA into the reaction cell led to a decrease in the $v$(C = O) peak at 1710 cm$^{-1}$, accompanied by the appearance of infrared peaks attributed to IPA (Fig. 4b). This suggests that both IPA and CAL

**Table 2 | Selective hydrogenation of cinnamaldehyde and hydrodeoxygenation of cinnamyl alcohol over various catalysts**

| Entry | Catalysts | Step 1ᵃ Conv.1a (%) | Y.1c (%) 1c | Step 2ᵇ Conv.1c (%) 1e | Y. (%) 1d | 1e+1f | 1g |
|---|---|---|---|---|---|---|---|
| 1 | None | 16.8 | – | 28.9 | – | – | – |
| 2 | ZIF-8 | 47.3 | 12.5 | 19.6 | – | – | – |
| 3 | ZnNC-500 | 29.4 | 3.7 | 13.8 | – | – | – |
| 4 | ZnNC-600 | 92.5 | 63.8 | 13.2 | 1.9 | 2.5 | – |
| 5 | ZnNC-700 | 30.4 | 6.5 | 33.8 | 1.7 | 18.5 | – |
| 6 | ZnNC-800 | 54.1 | 37.0 | 87.5 | 11.2 | 66.8 | – |
| 7 | ZnNC-900 | 95.0 | 78.2 | 99.9 | 11.0 | 75.7 | 4.6 |
| 8 | ZnNC-1000 | 41.4 | 16.8 | 57.4 | 12.3 | 26.8 | – |
| 9 | Zn(NO₃)₂ | 31.8 | 3.0 | 20.7 | – | – | – |
| 10 | ZnO | 36.5 | 8.6 | 0 | – | – | – |
| 11 | NC-900 | 16.8 | – | 0 | – | – | – |

Reaction conditions: ᵃCinnamaldehyde (0.2 mmol), catalyst (20 mg), 2-propanol (4 mL), $N_2$ (1 MPa), 150 °C, 11 h.
ᵇCinnamyl alcohol (0.2 mmol), catalyst (20 mg), 2-propanol (4 mL), $N_2$ (1 MPa), 180 °C, 7 h. The conversion of substrate and the yield of products were determined by GC with dodecane as internal standard.

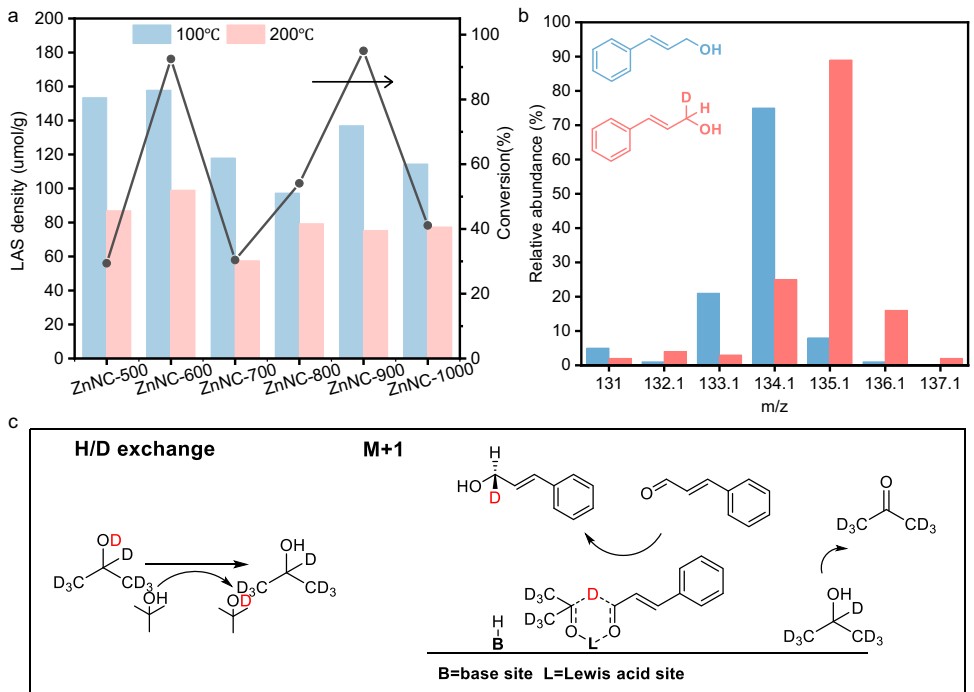

**Fig. 3 | Reaction mechanism. a** The relationship between Lewis acidic site (LAS) density of ZnNC-X catalysts and the conversion of cinnamaldehyde hydrogenation to cinnamyl alcohol. **b** Mass fragmentation analysis of the products. The blue is the mass spectrum of pure cinnamyl alcohol; the red is the mass spectrum of the reaction product. Reaction conditions: cinnamaldehyde (0.2 mmol), ZnNC-900 (20 mg), 2-propanol-$d_8$ (1 mL), $t$-butanol, 150 °C, 11 h, $N_2$ (1 MPa). The molar ratio of 2-propanol-$d_8$ to $t$-butanol is 1:3. **c** Reaction mechanism for the selective hydrogenation of cinnamaldehyde to cinnamyl alcohol.

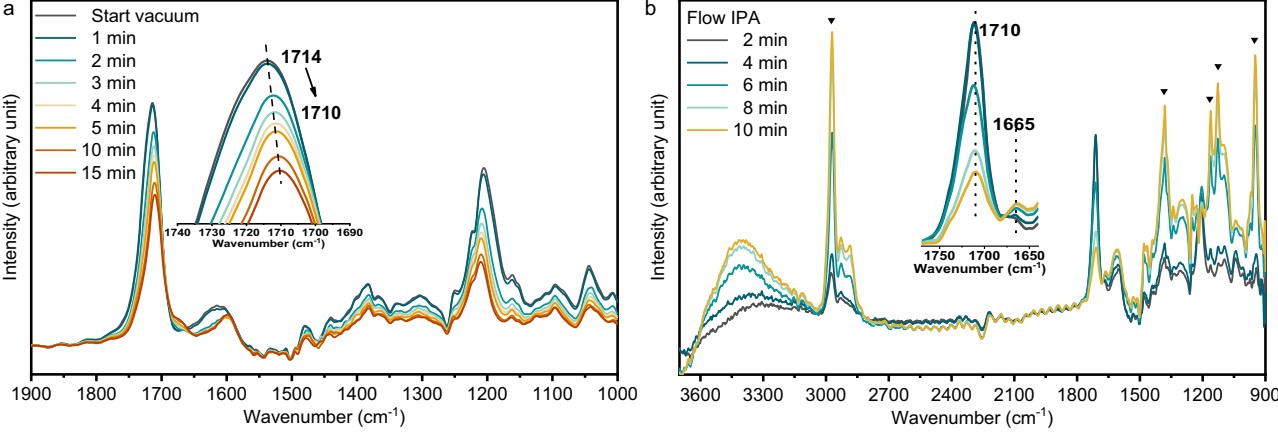

**Fig. 4 | In situ Fourier-transformed infrared spectra (FTIR). a** FTIR spectra of CAL adsorption on ZnNC-600 catalyst. **b** FTIR spectra of CAL and IPA adsorption on ZnNC-600 catalyst (The infrared peak at the inverted triangle represents part of the characteristic infrared peak of isopropyl alcohol.).

were adsorbed at the same site on the catalyst. Furthermore, a peak at 1665 cm$^{-1}$ was associated with the stretching vibration of the C = C bond[41,42]. Upon the introduction of IPA, the intensity of the $\nu$(C = C) band gradually increased, indicating a reduction in bridge adsorption of the C = C group. Consequently, the presence of IPA affects the adsorption mode of CAL. When CAL is present alone, both the C = O and C = C groups can be adsorbed on the catalyst. However, in the presence of CAL and IPA, CAL primarily adsorbs on the catalyst through the C = O bond, specifically via $\eta 1$(O) adsorption[42]. This observation explains why the catalyst selectively hydrogenates C = O bonds rather than C = C bonds, aligning with the proposed mechanism discussed earlier.

Gaseous hydrogen was detected in the reaction gas (Supplementary Fig. 16), but no COL product was found when $t$-butanol was

used as the solvent and $H_2$ was present (Supplementary Table 2). These findings suggest that the main pathway for the COL formation is catalytic transfer hydrogenation.

To investigate the active site responsible for COL hydrodeoxygenation, the catalysts were characterised using X-ray photoelectron spectroscopy (XPS) and Auger electron spectroscopy (AES). In ZnNC-600 and ZnNC-1000 catalysts, the Zn LMM binding energy was measured at 499.0 eV and 497.6 eV, respectively (Fig. 5a). With the increase of calcined catalyst temperature, the binding energy shifted slightly to lower binding energy[43]. This indicated that there was a low-valence Zn (Zn$^{\delta+}$, $0 < \delta < 2$) in ZnNC-X, which might be due to the formation of Zn-N$_x$ species. The N 1$s$ XPS spectrum of ZnNC-X can be deconvoluted into five types of N, which include pyridinic-N, Zn-N$_x$, pyrrolic-N, graphitic-N, and oxidised-N (Fig. 5b)[35,44,45]. Although ZnNC-

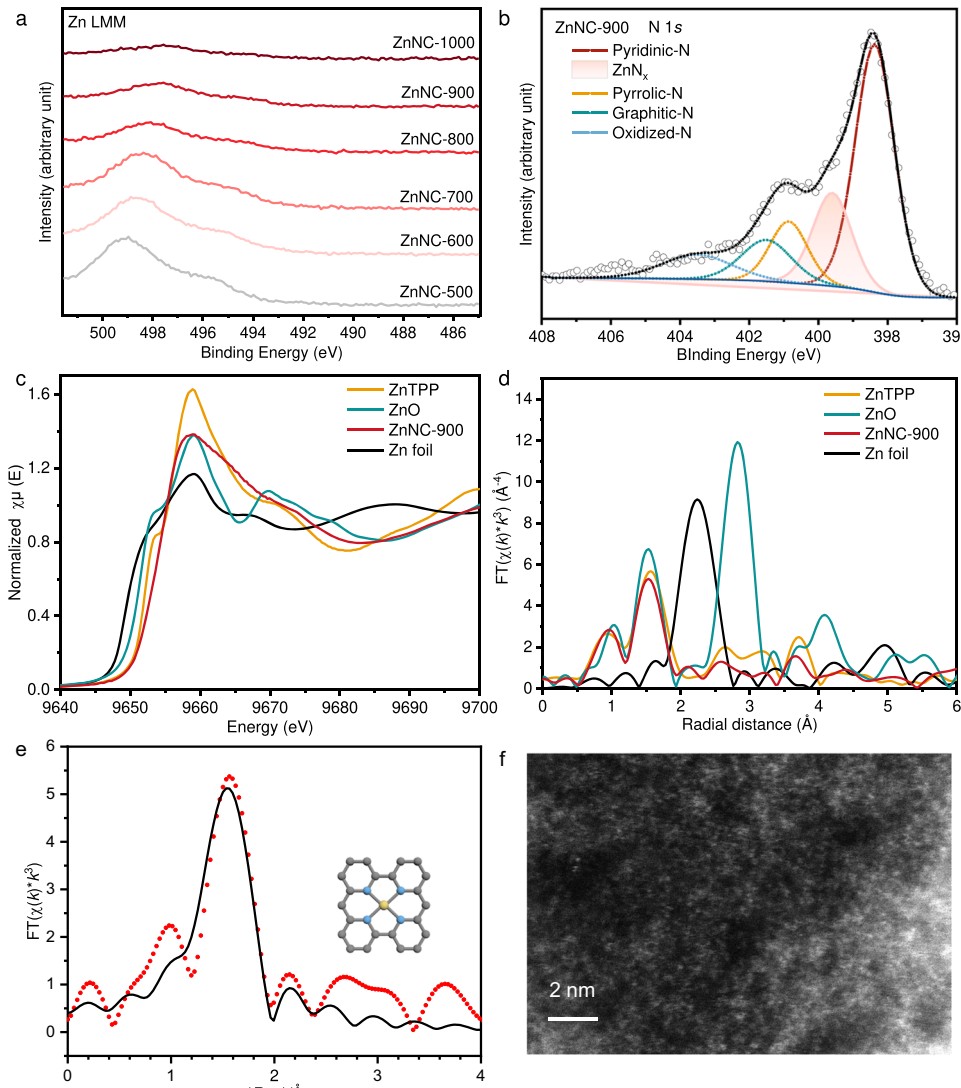

**Fig. 5 | The characterization of ZnNC-X catalysts. a** AES spectra of Zn LMM of Zn/NC-X catalysts. **b** XPS spectrum of N 1*s* of Zn/NC-900 catalyst. **c** Zn K-edge XANES spectra of ZnTPP, ZnO, ZnNC-900 catalyst and Zn foil. **d** Fourier transformed (FT) $k^3$-weighted Zn K-edge EXAFS spectra of ZnTPP, ZnO, ZnNC-900 catalyst and Zn foil. **e** Zn K-edge EXAFS (points) and the fitting curve (line) of ZnNC-900 catalyst, shown in R-space. The date is $k^3$-weighted and not phase-corrected (Zn is yellow, N is blue, C is grey). **f** AC-HAADF-STEM image of ZnNC-900 catalyst.

500 catalyst exhibited pyridinic-N with Lewis basicity, it did not show Lewis basicity in $CO_2$-TPD (Supplementary Fig. 12), possibly due to coordination of pyridinic-N to the Lewis acidic Zn site. Zn-$N_x$ species appeared at a pyrolysis temperature of 700 °C, with the highest content in ZnNC-900 catalyst (Supplementary Table 5). As the concentration of Zn-$N_x$ species increased, catalytic hydrodeoxygenation activity also increased. Therefore, it can be concluded that Zn-$N_x$ species serve as the active sites for hydrodeoxygenation, and ZnNC-900 catalyst exhibits the best catalytic performance for the selective hydrodeoxygenation of CAL to alkenes.

To determine the valence and coordination environment of Zn atom in the ZnNC-900 catalyst, X-ray absorption near-edge structure (XANES) and extended X-ray absorption fine structure (EXAFS) analyses were performed. Figure 5c shows that the white line of the ZnNC-900 catalyst fell between zinc tetraphenylporphyrin (ZnTPP) and zinc foil, indicating a valence state between 0 and +2. In the radial space (R space), no significant Zn–Zn bonding was detected at 2.27 Å, confirming the presence of atomically dispersed Zn in the catalyst (Fig. 5d)[35]. Aberration-corrected high-angle annular dark-field scanning transmission electron microscopy (AC-HAADF-STEM)

revealed isolated bright spots, further confirming the presence of individually dispersed zinc atoms (Fig. 5f). The Fourier transform (FT) $k^3$-weighted EXAFS spectrum of ZnNC-900 exhibited a main peak at approximately 1.93 Å, consistent with Zn-N coordination (Fig. 5e). The coordination number of N was determined to be 3.9 (Supplementary Table 7), which aligned with the findings from XPS and provided further evidence for the existence of Zn-$N_x$ species in the ZnNC-900 catalyst.

Control poison experiments were undertaken to ascertain the active sites involved in the hydrodeoxygenation of CAL to alkenes (Supplementary Table 4). In the initial step, it was observed that the presence of either base or acid led to a reduction in CAL conversion, emphasising the significance of Lewis acid-base sites for the reaction (Supplementary Table 4, Entry 2 and 3). KSCN, a recognised poison reagent capable of deactivating single atomic or metal sites[46], resulted in a substantial decrease in CAL conversion from 95.0% to 53.7% when 20 mg KSCN was added to the reaction (Supplementary Table 4, Entry 4). In the subsequent step, both pyridine and KSCN reduced the conversion of COL, while the impact of boric acid was minimal (Supplementary Table 4, Entry 6–8). These outcomes further indicate that the

**Table 3 | Selective hydrodeoxygenation of various α, β-unsaturated carbonyl compounds to alkenes**

| Entry | Substrate | Temp (°C) | Time (h) | Conversion (%) | Product 1 | Product 2 | Selectivity (%) |
|---|---|---|---|---|---|---|---|
| 1[a] | cinnamyl alcohol | 180 | 7 | >99.9 | (E)-propenylbenzene | allylbenzene | 75.7 |
| 2[a] | cinnamaldehyde | 180 | 24 | >99.9 | (E)-propenylbenzene | allylbenzene | 61.1 |
| 3[a] | (E)-benzalacetone | 190 | 12 | >99.9 | (E)-1-phenyl-1-butene | internal alkene | 75.5 |
| 4[b] | cinnamyl acetate | 180 | 7 | >99.9 | (E)-propenylbenzene | allylbenzene | 61.2 |
| 5[c] | 4-hydroxy-3-methoxycinnamaldehyde | 210 | 11 | >99.9 | propenyl product | allyl product | 67.5 |
| 6[c] | furan acrolein | 170 | 4 | >99.9 | furan propenyl | furan allyl | 94.8 |
| 7[c] | furan butenone | 170 | 17 | >99.9 | furan internal | furan internal | 81.7 |
| 8[c] | chalcone | 160 | 12 | >99.9 | 1,3-diphenylpropene | | 72.1 |

[a]Reaction conditions: substrate (0.2 mmol), ZnNC-900 (20 mg), 2-propanol (4 mL), N₂ (1 MPa), the conversion of substrate and the yield of products were determined by GC with dodecane as internal stansumidard.
[b]Substrate (10 mg).
[c]Area normalisation method was adopted to quantify products.

active site for the initial step was a Lewis acid-base site, whereas for the second step, the active sites consisted of $Zn-N_x$, as evidenced by the effect of KSCN on the reaction (Supplementary Table 4, Entry 8 and 12). In addition, the $Zn-N_x$ sites also served as the Lewis acid in the initial step.

Triphenylmethanol was selected as the substrate for studying the hydrodeoxygenation reaction mechanism. It is an ideal choice due to its hydrodeoxygenation producing a single product, triphenylmethane. Supplementary Fig. 18b demonstrates that the yield of triphenylmethane remained consistent even with varying amounts of triphenylmethanol. In addition, the conversion exhibited a linear relationship with time, indicating a zero-order reaction (Supplementary Fig. 19). The kinetic data in Supplementary Fig. 19 revealed a kinetic isotope effect (KIE) of 1.5 for the hydrodeoxygenation of triphenylmethanol, along with the detection of deuterated triphenylmethane. The study investigated the hydrogen source, revealing that 2-propanol could directly replace gaseous hydrogen based on the data in Supplementary Table 6. When KSCN was introduced to poison the reaction (Supplementary Table 6, Entry 4), a substantial decrease in the conversion rate of triphenylmethanol and the yield of triphenylmethane was observed, further confirming the involvement of the $Zn-N_x$ active site.

Based on the above experiments and in combination with relevant literature[23,47,48], we proposed a possible mechanism (Supplementary Fig. 23). Initially, IPA was adsorbed onto the surface of the ZnNC catalyst, interacting with Lewis acid-base sites. The hydroxyl O

of IPA was adsorbed at the Lewis acid ($Zn-N_x$) site, and the base site pyridine N lead to the dissociation of IPA. Subsequently, the carbonyl group of CAL adsorbed at this site and underwent the MPV reduction by forming a six-membered ring transition state. IPA was oxidised to acetone (ACE) and desorbed, and CAL was reduced to COL. Following this, another molecule of IPA attacked the allyl carbon on the adsorbed COL. The process possibly generated two ether intermediates through dehydration, which then underwent electron transfer, yielding terminal alkenes and internal alkenes catalysed by $Zn-N_x$ site.

## Substrate scope

The scope of hydrodeoxygenation catalysed by ZnNC-900 catalyst was demonstrated in Table 3. The conversion of substrates could reach 99.9%. The product selectivity reached 75.7% and 61.1% for cinnamyl alcohol and cinnamaldehyde, respectively (Table 3, Entry 1 and 2). For (E)-benzalacetone, the selectivity of alkenes was 75.5% at the higher temperature of 190 °C (Table 3, Entry 3). Cinnamyl acetate was also found to be reactive. The product selectivity was 61.2%, which was lower than that of cinnamyl alcohol (Table 3, Entry 4). This was probably due to the effect of hydrolysis of cinnamyl acetate to acetic acid. For 4-hydroxy-3-methoxycinnamaldehyde, one of the aldehyde monomers of lignin, 67.5% of the products was produced at higher temperatures (Table 3, Entry 5). ZnNC-900 catalyst could also catalyse unsaturated carbonyl compounds of furans (Table 3, Entry 6 and 7). And the reaction temperature of furan compounds was lower than that

of aromatic compounds. The products had a selectivity of 94.8% and 81.7% for (E)-3-(2-furanyl)-2-propenal and (E)-4-(furan-2-yl)but-3-en-2-one, respectively. For 1,3-diphenyl-propen-3-one, which contains two benzene rings, 1,3-diphenylpropene with a selectivity of 72.1% was formed at 160 °C (Table 3, Entry 8).

## Discussion

In conclusion, we have successfully achieved selective hydrodeoxygenation of α, β-unsaturated carbonyl compounds for the synthesis of olefinic compounds. The study shows that the reaction revealed a two-step reaction process involving the hydrogenation of α, β-unsaturated carbonyl compounds to produce unsaturated alcohols, followed by the hydrodeoxygenation of unsaturated alcohols to form olefins. We identified the active sites responsible for each step: the Lewis acid-base sites on the surface of ZnNC materials acted as the active sites for the initial hydrogenation reaction, while the Zn-N$_x$ species served as the active sites for the subsequent hydrodeoxygenation reaction. Moreover, we observed that ZnNC-900 catalyst exhibited broad applicability across a wide range of unsaturated carbonyl compounds. This work expands the scope of sustainable transformations in synthetic chemistry and contributes to the development of more efficient and environmentally friendly catalytic processes.

## Methods

### Materials

2-propanol-$d_8$ (99%), KBr (99.95%), (E)-benzalacetone, (E)-4-(furan-2-yl)but-3-en-2-one (98.0%), 1,3-diphenyl-propen-3-one (97%), 1-propenylbenzene (96%), 4-phenyl-1-butene were acquired from Shanghai Aladdin Biochemical Technology Co., Ltd. 2-methylimidazole (99%), 1-propanol (99.5%), (E)-3-(2-furanyl)-2-propenal (98%), cinnamyl alcohol (98%) were purchased from J&K Scientific Ltd. Methanol (99.7%), ethanol (99.7%), 2-propanol (99.8%), pyridine (99.5%), benzyl alcohol were acquired from Concord Technology (Tianjin) Co., Ltd. KSCN (99.5%), (E)-1-phenyl-1-butene (97%), triphenylmethane (99%), cinnamaldehyde (98%) were purchased from Shanghai Macklin Biochemical Co., Ltd. 1-butanol (99.5%), triphenylmethanol (99%), dodecane (99%), boric acid (99.5%), Zn(NO$_3$)$_2$·6H$_2$O, propylbenzene (99%), allylbenzene (98%), chitosan (99%) were acquired from Beijing InnoChem Science & Technology Co., Ltd. Cinnamyl acetate (97%) was acquired from TCI (Shanghai) Development Co., Ltd. 3-phenyl-1-propanol (98%) was purchased from Adamas-Beta. 4-hydroxy-3-methoxycinnamaldehyde (97%) was acquired from Sigma-Aldrich. t-butanol was purchased from Alfa Aesar. Nitrogen (99.99%), argon (99.99%) and hydrogen (99.99%) were provided by Beijing Analytic Instrument Company. All reagents were directly used without further purification.

### The preparation of Zn/NC-X catalysts

The ZnNC-X catalysts were prepared by the pyrolysis of ZIF-8[34]. Typically, 3.4 g Zn(NO$_3$)$_2$ and 3.7 g 2-methylimidazole were dissolved in 120 mL methanol, respectively. Then 2-methylimidazole/methanol solution was added to Zn(NO$_3$)$_2$/methanol solution under agitation. After stirring at 25 °C for 12 h, the white suspension was obtained. Then the suspension was centrifuged to obtain the white solid, which was washed twice with methanol and dried overnight in a 70 °C vacuum oven. The obtained sample was ground in an agate mortar and placed in a porcelain boat. The sample was then pyrolyzed at target temperature (500–1000 °C) for 3 h under argon atmosphere (The heating rate is 5 °C·min$^{-1}$, and the argon purge rate is 60 ml/min.). Finally, the catalyst was ground with an agate mortar.

### The preparation of NC-900 catalyst

The NC-900 catalyst was prepared by the pyrolysis of chitosan. Typically, chitosan was pyrolyzed at 900 °C for 3 h under argon atmosphere (The heating rate is 5 °C·min$^{-1}$, and the argon purge rate is

60 ml/min). Finally, the resulting black solid obtained by calcination was ground with an agate mortar.

### Catalytic activity tests

The reaction was carried out in a quartz-lined-stainless steel reactor with a capacity of 16 ml and a magnetic agitator. In a typical experiment, the required amount of catalyst, reactant, and solvent was added in sequence to the reactor. The air in the reactor was replaced by N$_2$ or H$_2$ three times. The reactor was then pressurised to a certain pressure with N$_2$ or H$_2$. The reactor was placed in the heating furnace and preheated for 1 h, so that the liquid in the reactor reached the required temperature. After a certain reaction time, the reactor was put into cold water, cooled to room temperature. Then a certain amount of internal standard was added to the reactor, and the reaction solution was diluted with a certain solvent. Then the catalyst was separated by centrifugation and the reaction liquid was analysed by gas chromatography. The conversion of reactants and the selectivity and yield of products were calculated based on gas chromatography data.

### Characterization

The scanning electron microscopy (SEM) measurements were measured on a Hitachi S-4800 scanning electron microscope, which operated at 15 kV. Transmission electron microscopy (TEM) images were obtained using a JEOL JEM-2010 transmission electron microscope or a Hitachi HT 7700 transmission electron microscope. The element energy dispersive spectroscopy (EDS) mappings were conducted on a JEOL JEM-2100F transmission electron microscope. Aberration-corrected high-angle annular dark-field scanning transmission electron microscopy (AC-HAADF-STEM) images were obtained on a JEM-ARM300F in situ aberration-corrected transmission electron microscopy. Powder X-ray diffraction (XRD) patterns were obtained by testing with Rigaku D/max-2500 X-ray diffractometer. Ray is Cu Kα Ray (λ = 0.154 nm), the operating voltage and current were 40 kV and 200 mA, respectively. N$_2$ adsorption-desorption isotherms were measured using Quantachrome Autosorb IQ. The content of zinc in ZnNC-X catalysts was measured by inductively coupled plasma optical emission spectrometer (ICP-OES) using Agilent ICP-OES 725 ES. The content of C, H and N elements in catalysts was measured by Thermo Flash Smart organic element analyser. CO$_2$ temperature programmed desorption (CO$_2$-TPD) and NH$_3$ temperature programmed desorption (NH$_3$-TPD) were obtained by Micromeritics/AUTOCHEM II 2920 chemisorption analyser. X-ray photoelectron spectroscopy (XPS) and Auger electron spectroscopy (AES) measurements were obtained from the Thermo Fisher Nexsa using Al Kα radiation. Fourier-transformed infrared spectra (FTIR) of the ZnNC-900 catalyst and NC-900 catalyst were measured by Bruker Vertex 70 V spectrometer. Fourier-transformed infrared spectra (FTIR) of the catalysts with pyridine absorbed on the catalysts were measured by Bruker TENSOR 27 spectrometer. In situ Fourier-transformed infrared spectra (FTIR) were measured by the FOLI10-R-T Fourier transform infrared spectrometer. The products were quantitatively analysed by the gas chromatography (Agilent 7890B) with a flame ionisation detector (FID). The gaseous products were analysed by the gas chromatography (Agilent 7890C) with a thermal conductivity detector (TCD). Qualitative analysis of the products was performed by gas chromatograph-mass spectrometer (Agilent 8860) with an electron impact (EI) ion source. Identification of the by-products was done using an orbitrap liquid chromatography-mass spectrometry (Thermo Scientific, Fusion Lumos) with an electrospray ionisation (ESI) ion source. $^1$H nuclear magnetic resonance (NMR) spectra were obtained on Bruker AV III 400 MHz NMR spectrometer. The X-ray absorption dates at the Zn K-edge of the samples were recorded at room temperature in transmission mode using ion chambers or in the fluorescent mode with silicon drift fluorescence detector at beam line BL11B of the Shanghai

Synchrotron Radiation Facility (SSRF), China. The station was operated with a Si (111) double crystal monochromator.

## Data availability
The primary data that support the plots within this paper and other finding of this study are available from the corresponding author on request.

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

## Acknowledgements

This work received financial support from National Key Research and Development Programme of China (2022YFA1504901) and the National Natural Science Foundation of China (22179132, 22121002, 22293012, 22293015, 22302209).

## Author contributions

T.J.W. performed most of the experiments and catalyst characterization. Y.X. carried out partial control experiments. B.Z., B.F.C., X.M.C. and Y.L. conducted part of the experimental investigation. M.H.D., Y.X.W. and S.L. assisted in the catalyst characterization. All authors contributed to discussions and participated in data analysis. T.J.W., H.Z.L. and B.X.H. co-wrote the manuscript with help from all co-authors. H.Z.L. and B.X.H. directed the research.

## Competing interests

The authors declare no competing interests.
