## [Peer Review File · Nature Communications]

Selective hydrodeoxygenation of α , β -unsaturated carbonyl compounds to alkenesREVIEWER COMMENTS

Reviewer #1 (Remarks to the Author):

This is an interesting paper that reports the catalytic conversion of unsaturated carbonyl compounds to alkenes on the ZnNC-X catalysts. The authors proposed a two-step mechanism in which the carbonyl group might be hydrogenated into hydroxyl group by the surface Lewis acid/base sites without affecting the C=C bond followed by the hydrodeoxygenation of the hydroxyl group to form an alkene on the Zn-Nx sites. Many characterization techniques were used to study the structures of catalysts and surface adsorption states. Generally speaking, the work was well designed and done and the paper was well organized and written.

However, there were still some drawbacks that need authors' attention.

(1) In Fig. 1 (C), the authors used decimal points for surface areas, which was not appropriate since the BET method has the theoretical error of 10%.

(2) The authors did not mention the roles played by carbon which was the most abundant element in their catalysts. Surface functional groups such as -COOH, -C=O and -OH must grow on carbon surfaces which are usually the acidic and basic sites which might play the roles for the hydrogenation of carbonyl to hydroxyl group via the CTH as the authors expected.

(3) The authors said that in the Zn-Nx, Zn cations were the Lewis acid sites while N the Lewis base sites, which were responsible for the hydrogenation of carbonyl to hydroxyl group. At the same time, the authors also demonstrated that the second step (the hydrodeoxygenation of hydroxyl group to alkenes) occurred on the Zn-Nx sites. Can the authors confirm that the two steps occurred on the same Zn-Nx sites (or distinguish them)?

(4) In Fig. 2, were there any evidences for the formation of the six-member ring containing D and the L site?

(5) There were some experimental evidences provided by the authors regarding the two-step mechanism. However, the catalyst surfaces were not involved in the mechanism. Hope that the authors may provide with some schematic diagrams depicting the surface structures of their catalysts in terms of Lewis acid/base and Zn-Nx sites interacted with adsorbed species showing the micro-processes for the two steps in the mechanism. I just did not get the point how the hydrodeoxygenation of hydroxyl group occurred on the Zn-Nx sites, neither the hydrogenation of carbonyl to hydroxyl group catalyzed by the Lewis acid/base sites.

(6) Some English grammar errors: (i) "To further clarify the important role of Lewis acid-base site for the catalytic selective hydrogenation. The control poison experiments were performed" (lines 119-120); (ii) "Interestingly, when the catalyst was poisoned with pyridine and boric acid, which targeted the acid and base sites, respectively. There was little impact on the catalytic performance (lines 200-202)". In each case, the two parts must be one sentence separate by a comma.

In view of above points, I suggest that a revision is necessary before the paper is considered for publication in Nature Communications.

Reviewer #2 (Remarks to the Author):

This paper describes an interesting catalytic process for producing alkenes from α , β -unsaturated carbonyl compounds via selective hydrodeoxygenation over a bifunctional catalyst. The authors carefully demonstrated that the Lewis acidic-basic sites were responsible for the selective hydrogenation of C=O bonds while the Zn-Nx species served as the active sites for the hydrodeoxygenation step. Because of this bifunctional mechanism, the ZnNC-900 catalyst showed a high catalytic activity in this challenging selective hydrodeoxygenation reaction. This work provided a new pathway for the production of olefins from biomass, which is significant for the development of green catalytic processes. Also, the bifunctional catalytic mechanism provided insights for designing efficient cascade catalysts. Therefore, this paper can be published in Nature Communications after addressing the following concerns.

1) Are there other catalysts reported previously for the transformation of α , β -unsaturated carbonyl compounds into alkenes via selective hydrodeoxygenation? The Introduction need to be updated for highlighting the advantages of the ZnNC-X catalyst containing two different active

sites and then presenting the importance of this work.

2) In Table 2, the yields are much lower than the conversions. Please add other products and their selectivities. It is important for understanding the reaction mechanism to analyze the reaction path.

3) Reaction kinetics of selective hydrodeoxygenation of cinnamaldehyde over all the ZnNC-X catalysts should be provided to better understand the reaction process proposed by the authors.

4) It is very interesting to construct two different active sites in a single support for sequential hydrogenation because it allows the efficient synthesis of value-added products. Relevant publications are listed: Nat. Commun. 12, 4968 (2021); Nat. Commun. 10, 4166 (2019); Nat. Mater. 15, 178–182 (2016).

5) The authors proposed that the Lewis acidic-basic sites were responsible for the selective hydrogenation of C=O bonds while the Zn-N_x species served as the active sites for the hydrodeoxygenation step. Are these two active sites compatible in the separate catalytic steps? Does the presence of the Zn-N_x species inhibit the selective hydrogenation of C=O bonds over the Lewis acidic-basic sites through competitive adsorption of reactants?

6) It seems that the Zn-N_x species also contributed to the selective hydrogenation of C=O bonds because ZnNC-1000 and ZnO also produced cinnamyl alcohol from cinnamaldehyde, although the yield was low. In the second step, the Lewis acidic-basic sites may also activate IPA to facilitate the hydrodeoxygenation of cinnamyl alcohol.

7) The format of the references is wrong and there are some mistakes in the manuscript. For example, in page 5 line 120, "CAL selectivity" should be "the conversion of CAL". The author is suggested to check through the paper.

Dear the editor and reviewers:

We would like to thank the editor and the reviewers for the positive and constructive comments regarding our paper. As you are concerned, there are several problems that need to be addressed. According to reviewers' useful suggestions, we have made corrections to our previous draft, the detailed corrections are listed below. Furthermore, we have resubmitted an updated manuscript, wherein the modifications are indicated in blue.

Reviewer #1 (Remarks to the Author):

This is an interesting paper that reports the catalytic conversion of unsaturated carbonyl compounds to alkenes on the ZnNC-X catalysts. The authors proposed a two-step mechanism in which the carbonyl group might be hydrogenated into hydroxyl group by the surface Lewis acid/base sites without affecting the C=C bond followed by the hydrodeoxygenation of the hydroxyl group to form an alkene on the Zn-N_x sites. Many characterization techniques were used to study the structures of catalysts and surface adsorption states. Generally speaking, the work was well designed and done and the paper was well organized and written. However, there were still some drawbacks that need authors' attention.

Response: We appreciate the favorable assessment of our work. The high evaluation from the reviewer is truly encouraging. Taking into account your constructive feedback and suggestions, we have revised our manuscript, thereby enhancing the quality of our work even further.

(1) In Fig. 1 (c), the authors used decimal points for surface areas, which was not appropriate since the BET method has the theoretical error of 10%.

Response: Thank you for pointing out this error, we have corrected it in the main text. (Please see Page 3 in the revised manuscript)

Figure 1. The structural characterization of ZnNC-X catalysts. **c** N₂ adsorption-desorption isotherms.

(2) The authors did not mention the roles played by carbon which was the most abundant element in their catalysts. Surface functional groups such as -COOH, -C=O and -OH must grow on carbon surfaces which are usually the acidic and basic sites which might play the roles for the hydrogenation of carbonyl to hydroxyl group via the CTH as the authors expected.

Response: Thanks for the reasonable comment. The NC-900 catalyst was prepared by pyrolysis of chitosan at 900°C under argon atmosphere directly. This catalyst did not contain zinc element, but contained C, N, O and H elements. We first tested the functional groups contained in the catalyst

using FTIR. “The peaks observed at 3447 cm^{-1} and 3592 cm^{-1} correspond to the stretching vibrations of the OH group. The peaks located at 1635 cm^{-1} and 1580 cm^{-1} are attributed to the stretching vibrations of the $\text{C}=\text{X}$ ($\text{X}=\text{C}$, O or N) bond.” (Please see Page 11 in the SI, marked in blue)

We then tested the catalytic activity of NC-900 and no products were detected as shown in Table 2 (Table 2, Entry 11). These findings suggest that the carbon-containing group may not serve as the catalytically active site. “Furthermore, it was established that the presence of zinc is essential for the reaction, as evidenced by the use of the NC-900 catalyst without zinc, which resulted in no product formation (Table 2, Entry 11).” (Please see Page 4 in the revised manuscript, marked in blue)

“**The preparation of NC-900 catalyst.** The NC-900 catalyst was prepared by the pyrolysis of chitosan. Typically, chitosan was pyrolyzed at 900 °C for 3 h under argon atmosphere (The heating rate is 5 °C·min⁻¹, and the argon purge rate is 60 ml/min.). Finally, the resulting black solid obtained by calcination was ground with an agate mortar.” (Please see Page 11 in the revised manuscript, marked in blue)

Table 2 Selective hydrogenation of cinnamaldehyde and hydrodeoxygenation of cinnamyl alcohol over various catalysts.

Entry	Catalysts	Step 1 ^a		Step 2 ^b			
		Conv. _{1a} (%)	Y. _{1c} (%)	Conv. _{1c} (%)	Y. (%)		
					1d	1e+1f	1g
11	NC-900	16.8	-	0	-	-	-

Reaction conditions: ^a cinnamaldehyde (0.2 mmol), catalyst (20 mg), 2-propanol (4 mL), N₂ (1 MPa), 150 °C, 11 h. ^b cinnamyl alcohol (0.2 mmol), catalyst (20 mg), 2-propanol (4 mL), N₂ (1 MPa), 180 °C, 7 h. The conversion of substrate and the yield of products were determined by GC with dodecane as internal standard

Supplementary Figure 9. FTIR spectra of ZnNC-900 catalyst and NC-900 catalyst.

(3) The authors said that in the Zn-N_x , Zn cations were the Lewis acid sites while N the Lewis base sites, which were responsible for the hydrogenation of carbonyl to hydroxyl group. At the same time, the authors also demonstrated that the second step (the hydrodeoxygenation of hydroxyl group to alkenes) occurred on the Zn-N_x sites. Can the authors confirm that the two steps occurred on the same Zn-N_x sites (or distinguish them)?

Response: The Zn-N_x sites also served as the Lewis acid in the initial step. It can be proved by control poison experiments and reaction kinetics.

“Control poison experiments were undertaken to ascertain the active sites involved in the hydrodeoxygenation of CAL to alkenes (Supplementary Table 4). In the initial step, it was observed that the presence of either base or acid led to a reduction in CAL conversion, emphasizing the significance of Lewis acid-base sites for the reaction (Supplementary Table 4, Entry 2 and 3). KSCN, a recognized poison reagent capable of deactivating single atomic or metal sites⁴⁶, resulted in a substantial decrease in CAL conversion from 95.0% to 53.7% when 20 mg KSCN was added to the reaction (Supplementary Table 4, Entry 4). In the subsequent step, both pyridine and KSCN reduced the conversion of COL, while the impact of boric acid was minimal (Supplementary Table 4, Entry 6-8). These outcomes further indicate that the active site for the initial step was a Lewis acid-base site, whereas for the second step, the active sites consisted of Zn-N_x, as evidenced by the effect of KSCN on the reaction (Supplementary Table 4, Entry 12). Additionally, the Zn-N_x sites also served as the Lewis acid in the initial step.” (Please see Page 8 in the revised manuscript, marked in blue)

“In the case of ZnNC-X (X=800-1000), as the reaction time extended, the yield of cinnamyl alcohol initially rose and then declined. Concurrently, the production of alkenes continued to increase, and the rate of alkene formation accelerated after nearly complete conversion of the CAL substrate. It could be seen from the time curves that the presence of CAL partially inhibited the hydrodeoxygenation of COL (Supplementary Fig. 7c-f). This indicated that CAL and COL competed for adsorption at the same site (Zn-N_x site).” (Please see Page 10 in the SI, marked in blue).

Supplementary Table 4. Effect of additives on reaction.

Entry	Substrates	Additives	Conv. (%)	Sel. (%)			
				1c	1d	1e+1f	1g
1 ^a	CAL	-	95.0	88.2	-	-	-
2 ^a	CAL	boric acid	70.3	52.5	-	-	-
3 ^a	CAL	pyridine	72.0	44.2	-	-	-
4 ^a	CAL	KSCN	53.7	36.8	-	-	-
5 ^b	COL	-	99.9	-	11.0	75.7	4.6
6 ^b	COL	boric acid	84.8	-	11.9	81.6	-
7 ^b	COL	pyridine	78.5	-	13.9	84.7	-
8 ^b	COL	KSCN	22.2	-	27.2	45.4	-
9 ^c	CAL	-	99.9	5.6	10.9	61.1	2.5
10 ^c	CAL	boric acid	99.9	18.5	10.1	55.8	-
11 ^c	CAL	pyridine	99.9	7.0	8.9	60.9	-
12 ^c	CAL	KSCN	99.9	54.0	9.5	14.1	-

Reaction conditions: substrates (0.2 mmol), ZnNC-900 (20 mg), IPA (4 mL), N₂ (1 MPa), poison reagents (20 mg). ^a 150°C, 11 h; ^b 180°C, 7 h; ^c 180 °C, 24 h. The conversion of substrate and the yield of products were determined by GC with dodecane as internal standard.

Supplementary Figure 7. Effect of reaction time of ZnNC-X catalysts.

46. Wang, R. Y. et al. Regulation of the Co-N_x active sites of MOF-templated Co@NC catalysts via Au doping for boosting oxidative esterification of alcohols. *ACS Catal.* **12**, 14290-14303 (2022).
 (4) In Fig. 2, were there any evidences for the formation of the six-member ring containing D and the L site?

Response: Thank you for the important comment. Deuteration experiments are generally used to demonstrate the Meerwein-Ponndorf-Verley (MPV) reduction mechanism, that is, the reaction passes through a six-membered ring mechanism. We are sorry that this issue may not be explained clearly enough in the text. Carbonyl compounds such as aldehydes and ketones are reduced to the corresponding alcohols in isopropyl alcohol with the catalyst, and isopropyl alcohol is oxidized to acetone, which usually involves the MPV reduction mechanism¹. The MPV reaction mechanism involves the simultaneous coordination of the carbonyl group and the hydroxy group to a metal center, as shown in the following picture³⁹. Román-Leshkov Y et al.³⁹ studied the mechanism of glucose isomerization by an intramolecular hydride shift, similar to MPV mechanism, which was proved by deuteration experiment. Gilkey, M. J. and Valekar, A. H. et al.^{2, 38} also performed deuteration experiments to demonstrate the concerted intermolecular hydride transfer mechanism (MPV).

We also demonstrated the mechanism through deuterium labeling experiments, establishing that the process involved Lewis acid-mediated intermolecular hydride transfer from the β -H in isopropyl alcohol to the carbonyl group. Furthermore, infrared spectroscopy indicated that aldehydes and isopropanol were adsorbed at the same site, providing further evidence for the MPV mechanism.

Editorial Note: Figure below reproduced from Román-Leshkov, Y., Moliner, M., Labinger, J.A. and Davis, M.E. (2010), Mechanism of Glucose Isomerization Using a Solid Lewis Acid Catalyst in Water. *Angewandte Chemie International Edition*, **49**: 8954-8957, <https://doi.org/10.1002/anie.201004689>, with permission from John Wiley & Sons. Copyright © 2010 WILEY-VCH Verlag GmbH & Co. KGaA, Weinheim.

Figure S1: Meerwein-Ponndorf-Verley (MPV) reaction pathway. R = alkyl or aryl; R₁ and R₂ = alkyl or hydrogen; Me = metal

“This means that β -D from 2-propanol- d_7 was directly transferred to CAL via a six-membered ring intermediate through Meerwein-Ponndorf-Verley (MPV) reduction^{38, 39} (Figure 2c).” (Please see Page 6 in the revised manuscript, marked in blue).

1. Corma, A., Domine, M. E., Nemeth, L. & Valencia, S. Al-free Sn-Beta zeolite as a catalyst for the selective reduction of carbonyl compounds (Meerwein-Ponndorf-Verley reaction). *J. Am. Chem. Soc.* **124**, 3194-3195 (2002).

2. Valekar, A. H. et al. Catalytic transfer hydrogenation of furfural to furfuryl alcohol under mild conditions over Zr-MOFs: exploring the role of metal node coordination and modification. *ACS Catal.* **10**, 3720-3732 (2020).

38. Gilkey, M. J. et al. Mechanistic insights into metal Lewis acid-mediated catalytic transfer hydrogenation of furfural to 2-methylfuran. *ACS Catal.* **5**, 3988-3994 (2015).

39. Román-Leshkov, Y., Moliner, M., Labinger, J. A. & Davis, M. E. Mechanism of glucose isomerization using a solid Lewis acid catalyst in water. *Angew. Chem. Int. Ed.* **49**, 8954-8957 (2010).

(5) There were some experimental evidences provided by the authors regarding the two-step mechanism. However, the catalyst surfaces were not involved in the mechanism. Hope that the authors may provide with some schematic diagrams depicting the surface structures of their catalysts in terms of Lewis acid/base and Zn-N_x sites interacted with adsorbed species showing the micro-processes for the two steps in the mechanism. I just did not get the point how the hydrodeoxidation of hydroxyl group occurred on the Zn-N_x sites, neither the hydrogenation of carbonyl to hydroxyl group catalyzed by the Lewis acid/base sites.

Response: We appreciate the question and apologize for not clearly explaining the mechanism that occurs on the surface of the catalyst. We drew a schematic to further clarify the mechanism (Supplementary Fig. 23). (Please see Page 21 in the SI, marked in blue).

“Based on the above experiments and in combination with relevant literature^{23, 47, 48}, we proposed a possible mechanism (Supplementary Fig. 23). Initially, isopropyl alcohol was adsorbed onto the surface of the ZnNC catalyst, interacting with Lewis acid-base sites. The hydroxyl O of isopropyl alcohol was adsorbed at the Lewis acid (Zn-N_x) site, and the base site pyridine N lead to the dissociation of isopropyl alcohol. Subsequently, the carbonyl group of CAL adsorbed at this site and underwent the MPV reduction by forming a six-membered ring transition state. IPA was oxidized to acetone (ACE) and desorbed, and CAL was reduced to COL. Following this, another molecule of isopropyl alcohol attacked the allyl carbon on the adsorbed cinnamyl alcohol. The process possibly generated two ether intermediates through dehydration, which then underwent

electron transfer, yielding terminal alkenes and internal alkenes catalyzed by Zn-N_x.” (Please see Page 8 in the revised manuscript, marked in blue).

Supplementary Figure 23. The possible reaction mechanism. (Zn is yellow. N is blue. C is gray.)

23. Fan, Y. F. et al. Efficient single-atom Ni for catalytic transfer hydrogenation of furfural to furfuryl alcohol. *J. Mater. Chem. A* **9**, 1110-1118 (2021).

47. Liu, W. G. et al. A durable nickel single-atom catalyst for hydrogenation reactions and cellulose valorization under harsh conditions. *Angew. Chem. Int. Ed.* **130**, 7189-7193 (2018).

48. Liu, Q. W. et al. Direct deoxygenation of active allylic alcohols via metal-free catalysis. *Org. Biomol. Chem.* **20**, 1680-1689 (2022).

(6) Some English grammar errors: (i) “To further clarify the important role of Lewis acid-base site for the catalytic selective hydrogenation. The control poison experiments were performed” (lines 119–120); (ii) “Interestingly, when the catalyst was poisoned with pyridine and boric acid, which targeted the acid and base sites, respectively. There was little impact on the catalytic performance (lines 200–202)”. In each case, the two parts must be one sentence separate by a comma.

Response: Thanks for your careful checks. We are sorry for our carelessness. Based on your comments, we have made the corrections.

In view of above points, I suggest that a revision is necessary before the paper is considered for publication in Nature Communications.

Reviewer #2 (Remarks to the Author):

This paper describes an interesting catalytic process for producing alkenes from α , β -unsaturated carbonyl compounds via selective hydrodeoxygenation over a bifunctional catalyst. The authors carefully demonstrated that the Lewis acidic-basic sites were responsible for the selective hydrogenation of C=O bonds while the Zn-N_x species served as the active sites for the hydrodeoxygenation step. Because of this bifunctional mechanism, the ZnNC-900 catalyst showed a high catalytic activity in this challenging selective hydrodeoxygenation reaction. This work provided a new pathway for the production of olefins from biomass,

which is significant for the development of green catalytic processes. Also, the bifunctional catalytic mechanism provided insights for designing efficient cascade catalysts. Therefore, this paper can be published in Nature Communications after addressing the following concerns.

Response: We are very grateful for your insightful and constructive comments, which we find very helpful in revising our manuscript.

1) Are there other catalysts reported previously for the transformation of α , β -unsaturated carbonyl compounds into alkenes via selective hydrodeoxygenation? The Introduction need to be updated for highlighting the advantages of the ZnNC-X catalyst containing two different active sites and then presenting the importance of this work.

Response: This is a helpful suggestion that reminds us to describe a more comprehensive research background. There are some relevant reports on the transformation of α , β -unsaturated carbonyl compounds into alkenes via selective hydrodeoxygenation. These references were cited in the text. Relevant publications about the transformation of α , β -unsaturated carbonyl compounds into alkenes are listed:

- Reference 11 *Tetrahedron letters*, 1995, 36(13): 2347-2350.
- Reference 12 *Chemical Communications*, 2018, 54(38): 4834-4837.
- Reference 13 *Chemistry of Heterocyclic Compounds*, 1980, 16: 339-344.
- Reference 14 *Tetrahedron*, 2000, 56(47): 9181-9193.
- Reference 15 *ACS Catalysis*, 2021, 11(21): 13337-13347.
- Reference 16 *Chemical Communications*, 1997 (17): 1647-1648.

We have added a description in the introduction, highlighting the advantages of the ZnNC-X catalyst containing two different active sites, and then presenting the importance of this work. Here is what was added in the introduction.

“Several literature sources have documented the synthesis of alkenes from α - β -unsaturated carbonyl compounds^{11, 12}. For instance, this has been achieved through reduction via hydrazone intermediates in the Wolff-Kishner-Huang reduction¹³ or through xanthate intermediates in the Barton-McCombie reaction¹⁴. Cook A et al. reported its accomplishment using a Ni(II) pre-catalyst and a silane reducing agent¹⁵, while Gómez A M et al. demonstrated the process in two reaction steps¹⁶. However, the reagents used in these methods are not considered safe or environmentally friendly. To facilitate one-pot cascade reactions, the development of bifunctional catalysts is essential. Various nanostructured materials such as multicompartmentalized mesoporous organosilicas¹⁷, nanotubes¹⁸, and hierarchical architectures¹⁹ have been explored to fabricate efficient cascade catalysts. Bifunctional catalysts can be created by loading bimetallic components onto these nanostructured materials, thereby enabling effective catalysis through synergistic interactions between multiple sites.” (Please see Pages 1 and 2 in the revised manuscript, marked in blue)

11. Srikrishna, A., Viswajanani, R., Sattigeri, J. A. & Yelamaggad, C. V. Chemoselective reductive deoxygenation of α , β -unsaturated ketones and allyl alcohols. *Tetrahedron Lett.* **36**, 2347-2350 (1995).

12. Yang, W. Y., Gao, L., Lu, J. & Song, Z. L. Chemoselective deoxygenation of ether-substituted alcohols and carbonyl compounds by B(C₆F₅)₃-catalyzed reduction with (HMe₂SiCH₂)₂. *Chem. Commun.* **54**, 4834-4837 (2018).

13. Mochalov, S. S., Abdel'razek, F. M., Surikova, T. P. & Shabarov, Y. S. Compounds nitration of 5-formyl-substituted 2-cyclopropylfurans and 2-methylfurans and the corresponding thiophenes. *Chem. Heterocycl. Compds.* **16**, 339-344 (1980).
14. Nascimento, I. R., Lopes, L. M. X., Davin, L. B. & Lewis, N. G. Stereoselective synthesis of 8,9-licarinediols. *Tetrahedron* **56**, 9181-9193 (2000).
15. Cook, A., MacLean, H., St. Onge, P. & Newman, S. G. Nickel-catalyzed reductive deoxygenation of diverse C-O bond-bearing functional groups. *ACS Catal.* **11**, 13337-13347 (2021).
16. Gómez, A. M., López de Uralde, B., Valverde, S. & Cristóbal López, J. A novel entry to naturally occurring 5-alkenyl α , β -unsaturated δ -lactones from D-glucose: syntheses of (+)-acetylphomalactone and (+)-asperlin. *Chem. Commun.* **17**, 1647-1648 (1997).
17. Zou, H. B. et al. Dual metal nanoparticles within multicompartimentalized mesoporous organosilicas for efficient sequential hydrogenation. *Nat. Commun.* **12**, 4968 (2021).
18. Zhang, J. K. et al. Origin of synergistic effects in bicomponent cobalt oxide-platinum catalysts for selective hydrogenation reaction. *Nat. Commun.* **10**, 4166 (2019).
19. Parlett, C. M. A. et al. Spatially orthogonal chemical functionalization of a hierarchical pore network for catalytic cascade reactions. *Nat. Mater.* **15**, 178-182 (2015).

2) In Table 2, the yields are much lower than the conversions. Please add other products and their selectivities. It is important for understanding the reaction mechanism to analyze the reaction path.

Response: Thank you for the important comment. We regret very much that we cannot give selectivity for other products but we explain why the yield is low. The first reason is that the raw material of cinnamic aldehyde is easily deteriorated and thus impured; it is easily oxidized to cinnamic acid or decomposed into benzaldehyde and carbon dioxide. In addition, cinnamaldehyde was polymerized during the reaction. These two points were the main reasons for the poor carbon balance. Some of the byproducts were detected by GC-MS and LC-MS. These byproducts were mainly produced by the aldol condensation of cinnamaldehyde and acetone.

“(E, E)-6-phenyl-3,5-hexadien-2-one, one of the by-products, was obtained by aldol condensation of cinnamaldehyde with acetone and then dehydration. (Supplementary Figure 10)” “The by-products may be formed by aldol condensation of one molecule of acetone and two molecules of cinnamaldehyde. (Supplementary Figure 11)” (Please see Pages 12 and 13 in the SI, marked in blue)

Supplementary Figure 10. Mass spectrum of the by-product by GC-MS (EI).

Supplementary Figure 11. Mass spectra of the by-products by LC-MS (ESI).

a C₂₁H₂₂O₃ LC-MS (ESI): m/z calcd for C₂₁H₂₃O₃ [M+H]⁺:323.1641; found: 323.1267.

b C₂₁H₁₈O LC-MS (ESI): m/z calcd for C₂₁H₁₈ONa [M+Na]⁺:309.1249; found: 309.2072.

c C₂₁H₂₀O LC-MS (ESI): m/z calcd for C₂₁H₂₁O [M+H]⁺: 289.1587; found: 289.1597.

3) Reaction kinetics of selective hydrodeoxygenation of cinnamaldehyde over all the ZnNC-X catalysts should be provided to better understand the reaction process proposed by the authors.

Response: Thanks for your valuable advice, this kinetic experiment has inspired us to further understand the reaction process.

“Supplementary Figure 7 displays the time curves for ZnNC-X (X=500-1000). It is evident from the data that the primary product catalyzed by ZnNC-X (X=500-600) was cinnamyl alcohol. In the case of ZnNC-X (X=800-1000), as the reaction time extended, the yield of cinnamyl alcohol initially rose and then declined. Concurrently, the production of alkenes continued to increase, and the rate of alkene formation accelerated after nearly complete conversion of the CAL substrate. It could be seen from the time curves that the presence of CAL partially inhibited the hydrodeoxygenation of COL (Supplementary Fig. 7c-f).” (Please see Page 10 in the SI, marked in blue)

Supplementary Figure 7. Effect of reaction time of ZnNC-X catalysts. a ZnNC-500. b ZnNC-600. c ZnNC-700. d ZnNC-800. e ZnNC-900. f ZnNC-1000. Reaction conditions: cinnamaldehyde (0.2 mmol), ZnNC-X (20 mg), 2-propanol (4 mL), 180 °C, N₂ (1 MPa).

4) It is very interesting to construct two different active sites in a single support for sequential hydrogenation because it allows the efficient synthesis of value-added products. Relevant publications are listed: Nat. Commun. 12, 4968 (2021); Nat. Commun. 10, 4166 (2019); Nat. Mater. 15, 178–182 (2016).

Response: Thank you for providing us with these important articles and we have carefully read them. The important recent literatures have been cited as reference 17, 18 and 19 in our revised manuscript. We will continue to focus on the development of these areas in future research. (Please see Pages 13 and 14 in the revised manuscript, marked in blue)

5) The authors proposed that the Lewis acidic-basic sites were responsible for the selective hydrogenation of C=O bonds while the Zn-N_x species served as the active sites for the hydrodeoxygenation step. Are these two active sites compatible in the separate catalytic steps? Does the presence of the Zn-N_x species inhibit the selective hydrogenation of C=O bonds over the Lewis acidic-basic sites through competitive adsorption of reactants?

Response: Thanks for the comment. The Zn-N_x sites also served as the Lewis acid in the initial step. So the presence of the Zn-N_x species do not inhibit the selective hydrogenation of C=O bonds. Evidences that Zn-N_x sites also served as the Lewis acid on ZnNC-900 catalyst are described below.

“Control poison experiments were undertaken to ascertain the active sites involved in the hydrodeoxygenation of CAL to alkenes (Supplementary Table 4). In the initial step, it was observed that the presence of either base or acid led to a reduction in CAL conversion, emphasizing the significance of Lewis acid-base sites for the reaction (Supplementary Table 4, Entry 2 and 3). KSCN, a recognized poison reagent capable of deactivating single atomic or metal sites⁴⁶, resulted in a

substantial decrease in CAL conversion from 95.0% to 53.7% when 20 mg KSCN was added to the reaction (Supplementary Table 4, Entry 4). In the subsequent step, both pyridine and KSCN reduced the conversion of COL, while the impact of boric acid was minimal (Supplementary Table 4, Entry 6-8). These outcomes further indicate that the active site for the initial step was a Lewis acid-base site, whereas for the second step, the active sites consisted of Zn-N_x, as evidenced by the effect of KSCN on the reaction (Supplementary Table 4, Entry 12). Additionally, the Zn-N_x sites also served as the Lewis acid in the initial step.” (Please see Page 8 in the revised manuscript, marked in blue)

“In the case of ZnNC-X (X=800-1000), as the reaction time extended, the yield of cinnamyl alcohol initially rose and then declined. Concurrently, the production of alkenes continued to increase, and the rate of alkene formation accelerated after nearly complete conversion of the CAL substrate. It could be seen that the presence of CAL partially inhibited the hydrodeoxygenation of COL (Supplementary Fig. 7c-f). This indicated that CAL and COL competed for adsorption at the same site (Zn-N_x site).” (Please see Page 10 in the SI, marked in blue).

Supplementary Table 4. Effect of additives on reaction.

Entry	Substrates	Additives	Conv. (%)	Sel. (%)			
				1c	1d	1e+1f	1g
1 ^a	CAL	-	95.0	88.2	-	-	-
2 ^a	CAL	boric acid	70.3	52.5	-	-	-
3 ^a	CAL	pyridine	72.0	44.2	-	-	-
4 ^a	CAL	KSCN	53.7	36.8	-	-	-
5 ^b	COL	-	99.9	-	11.0	75.7	4.6
6 ^b	COL	boric acid	84.8	-	11.9	81.6	-
7 ^b	COL	pyridine	78.5	-	13.9	84.7	-
8 ^b	COL	KSCN	22.2	-	27.2	45.4	-
9 ^c	CAL	-	99.9	5.6	10.9	61.1	2.5
10 ^c	CAL	boric acid	99.9	18.5	10.1	55.8	-
11 ^c	CAL	pyridine	99.9	7.0	8.9	60.9	-
12 ^c	CAL	KSCN	99.9	54.0	9.5	14.1	-

Reaction conditions: substrates (0.2 mmol), ZnNC-900 (20 mg), IPA (4 mL), N₂ (1 MPa), poison reagents (20 mg). ^a 150°C, 11 h; ^b 180°C, 7 h; ^c 180 °C, 24 h. The conversion of substrate and the yield of products were determined by GC with dodecane as internal standard.

Supplementary Figure 7. Effect of reaction time of ZnNC-X catalysts.

6) It seems that the Zn-N_x species also contributed to the selective hydrogenation of C=O bonds because ZnNC-1000 and ZnO also produced cinnamyl alcohol from cinnamaldehyde, although the yield was low. In the second step, the Lewis acidic-basic sites may also activate IPA to facilitate the hydrodeoxygenation of cinnamyl alcohol.

Response: Thanks for the comment and we agree with the comment. Zn-N_x, as a Lewis acid site, can catalyze C=O hydrogenation. We proposed a possible mechanism, as shown in Supplementary Figure 23. In the second step, the Lewis acidic-basic sites may also activate IPA to facilitate the hydrodeoxygenation of cinnamyl alcohol. “In the subsequent step, both pyridine and KSCN reduced the conversion of COL, while the impact of boric acid was minimal (Supplementary Table 4, Entry 6-8).” “For the second step, the active sites mainly consisted of Zn-N_x, as evidenced by the effect of KSCN on the reaction.” (Please see Pages 8 line 214 in the revised manuscript, marked in blue).

“Based on the above experiments and in combination with relevant literature^{23,47,48}, we proposed a possible mechanism (Supplementary Fig. 23). Initially, isopropyl alcohol was adsorbed onto the surface of the ZnNC catalyst, interacting with Lewis acid-base sites. The hydroxyl O of isopropyl alcohol was adsorbed at the Lewis acid (Zn-N_x) site, and the base site pyridine N lead to the dissociation of isopropyl alcohol. Subsequently, the carbonyl group of CAL adsorbed at this site and underwent the MPV reduction by forming a six-membered ring transition state. IPA was oxidized to acetone (ACE) and desorbed, and CAL was reduced to COL. Following this, another molecule of isopropyl alcohol attacked the allyl carbon on the adsorbed cinnamyl alcohol. The process possibly generated two ether intermediates through dehydration, which then underwent electron transfer, yielding terminal alkenes and internal alkenes catalyzed by Zn-N_x.” (Please see Page 8 in the revised manuscript, marked in blue).

Supplementary Figure 23. The possible reaction mechanism. (Zn is yellow. N is blue. C is gray.)

23. Fan, Y. F. et al. Efficient single-atom Ni for catalytic transfer hydrogenation of furfural to furfuryl alcohol. *J. Mater. Chem. A* **9**, 1110-1118 (2021).

47. Liu, W. G. et al. A durable nickel single-atom catalyst for hydrogenation reactions and cellulose

valorization under harsh conditions. *Angew. Chem. Int. Ed.* **130**, 7189-7193 (2018).

48. Liu, Q. W. et al. Direct deoxygenation of active allylic alcohols via metal-free catalysis. *Org. Biomol. Chem.* **20**, 1680-1689 (2022).

7) The format of the references is wrong and there are some mistakes in the manuscript. For example, in page 5 line 120, “CAL selectivity” should be “the conversion of CAL”. The author is suggested to check through the paper.

Response: Thank you for your careful review of our manuscript. We were sorry for our careless mistakes. We have corrected the format of the references and revised the sentence in the main text.

REVIEWERS' COMMENTS

Reviewer #1 (Remarks to the Author):

I saw that the comments from the reviews have been seriously addressed by the authors and the paper has been significantly improved. Now it can be accepted as it is.

Reviewer #2 (Remarks to the Author):

This revised manuscript has been significantly improved and all my concerns have been well addressed. This paper could be published.